# Fault Diagnosis of Loader Gearbox Based on an ICA and SVM Algorithm

**DOI:** 10.3390/ijerph16234868

**Published:** 2019-12-03

**Authors:** Zhongxin Chen, Feng Zhao, Jun Zhou, Panling Huang, Xutao Zhang

**Affiliations:** 1School of Mechanical Engineering, Shandong University, Jinan 250061, China; 201920453@mail.sdu.edu.cn (Z.C.); zhaofeng@mail.sdu.edu.cn (F.Z.); zhangxutao@mail.sdu.edu.cn (X.Z.); 2Key Laboratory of High Efficiency and Clean Mechanical Manufacture, Ministry of Education, Jinan 250061, China

**Keywords:** independent component analysis, support vector machines, feature extraction, fault diagnosis, gearbox

## Abstract

When a part of the loader’s gearbox fails, this can lead to equipment failure due to the complex internal structure and the interrelationship between the parts. Therefore, it is imperative to research an efficient strategy for transmission fault diagnosis. In this study, the non-contact characteristics of noise diagnosis using sound intensity probes were used to collect noise signals generated under gear breaking conditions. The independent component analysis (ICA) technique was applied for feature extraction from the original data and to reduce the correlation between the signals. The correlation coefficient between the independent components and the source data was used as the input parameters of the support vector machine (SVM) classifier. The separation of the independent components was achieved by MATLAB simulation. The misdiagnosis rate was 5% for 40 sets of test data. A 13-point test platform for noise testing of the loader gearbox was built according to Chinese national standards. Source signals under the normal and fault conditions were analyzed separately by ICA and SVM algorithms. In this case, the misdiagnosis rate was 7.5% for the 40 sets of experimental test data. This proved that the proposed method could effectively realize fault classification and recognition.

## 1. Introduction

As the most widely used type of engineering machinery, a loader is commonly utilized in construction sites such as roads, railways, and ports. Due to poor working conditions, it is particularly important to detect the failure of this construction machinery, especially through online detection. The gearbox is an important transmission component in loaders. Statistics show that the gear is the most common part causing failure: approximately sixty percent of failures are caused by the gear [1]. At present, the diagnosis methods for gearbox faults include temperature detection, optical fiber sensing, and vibration detection. These methods require that the position of the measuring instrument be changed when examining different fault locations at the same or different gearboxes. This is time-consuming and inefficient. Therefore, it is particularly important to propose a reasonable method for fault diagnosis. 

Independent component analysis (ICA) is a new signal analysis technology developed in recent decades from the blind signal separation problem. It has been successfully applied in fields such as bioengineering [2], communication [3], speech recognition [4], and fault diagnosis [5]. In the field of neuroimaging, Melissant et al. used the ICA algorithm to preprocess the EEGs of patients with Alzheimer’s disease, which improved the accuracy of diagnosis [6]. Gohel effectively repressed signal leakage by using the orthogonality of the ICA to reconstruct the source signal [7]. Back was the first to introduce ICA to the field of financial analysis, and many scholars have since used ICA to analyze and process financial data [8]. Independent component analysis was applied to the financial time series data belonging to a retail chain, where it detected a few factors that affect the cash flow [9]. Independent component analysis has also been used in machine condition monitoring and fault diagnosis. For example, an approach involving dynamic independent component analysis in fault detection and diagnosis, proposed by George, is able to accurately detect and isolate the root causes for each individual fault [10]. Principal component analysis (PCA) and ICA techniques were applied to the diagnosis of turbines for fault detection, and the validity and effectiveness of the approach were verified [11]. The French scholar Badaoui used ICA in combination with a Wiener filter to separate combustion noise and piston slap in a diesel engine [12]. Many improved algorithms such as constrained independent component analysis (cICA) [13,14] and kernel independent component analysis (KICA) [15,16] have been developed in order to improve the practical range and precision of the ICA algorithm. 

The application of an intelligent system for fault diagnosis has been widely researched in recent years. Support vector machines (SVMs) and neural networks have been employed to solve classification problems [17]. In the field of economics, Jae identified the optimal parameter values of the kernel function of the SVM to predict bankruptcy and obtained better performance than multiple discriminant analysis (MDA), Logit, and back-propagation neural networks (BPNs) [18]. Linear SVM classifiers were introduced for credit risk evaluation [19]. In signal processing, Salvati used an SVM to locate near-field sound sources and improved the accuracy of the beam former in a reverberation environment [20]. Haik built an optimized SVM-based probability model to select a window from a continuous data, and the effectiveness of the approach was proven [21]. Many scholars regard SVM as an intelligent classifier for machine fault classification, especially bearing faults [22,23], gear faults [24,25], wind turbine faults [26,27], and induction motors [28,29]. When applied in the current study, SVM has high computational complexity and low precision when directly classifying using raw data with redundant characteristics. Therefore, the ICA technique was applied for feature extraction from the original data and to reduce the correlation between the signals. The correlation coefficients between the feature vector and the source data were then used as the input parameters of the SVM classifier.

## 2. Fault Diagnosis Algorithm

### 2.1. Independent Component Analysis

Independent component analysis is a technique that has been demonstrated to be capable of separating an independent component from the source signal by utilizing its independence and non-Gaussian property. The process used by ICA is illustrated in Figure 1.

In Figure 1, the source signal is composed of a few independent acoustic sources, $s_{n}\left(t\right)$. Considering the effect of path delay in the signal transmission process, each signal can take multiple channels to reach the receivers, $x_{m}\left(t\right)$. The process of unmixing the signals is performed by the ICA algorithm. Assuming that n known independent source signals were transmitted by an unknown channel and eventually accepted by m receivers, there are n independent components $y_{n}\left(t\right)$, which constitute the estimates of the source signal. The received signal can be expressed as [30]
(1)$$x_{i}(t)=\overset{n}{\underset{j=1}{\sum }}\overset{P}{\underset{p=0}{\sum }}b_{ij}(p)s_{j}(t-p)$$
where $x_{i}\left(t\right)$ is the mixed signal received by receiver *i*, $b_{ij}$ is the response function between the signal from the *i*th receiver and the *j*th source signal, and $s_{j}\left(t-p\right)$ is the delay function from the *j*th source signal to the *i*th receiver. Its matrix form is
(2)X(t)=B(t)∗S(t)
where B(t) is a mixed matrix in the time domain.

Equation (2) is subjected to Fourier transform:(3)X(f)=B(f)S(f)
where $X\left(f\right)={\left[x_{1}\left(f\right),x_{2}\left(f\right),L,x_{m}\left(f\right)\right]}^{T}$ is the observed signal matrix that has been transformed in the frequency domain, b(f) is the m×n hybrid matrix, and $S\left(f\right)={\left[s_{1}\left(f\right),s_{2}\left(f\right),L,s_{n}\left(f\right)\right]}^{T}$ is the source signal matrix. In the processing of the mixed solution, a linear matrix W is determined to obtain the estimated signal from the source signal. Therefore, the separation of the ICA is given by
(4)Y(f)=WX(f)=WAS(f)
where Y(f) is the estimation of S(f) in the frequency domain, and the inverse Fourier transform is used to transform the frequency to the time domain, *A* is an unknown mixing matrix.
(5)$$A={\left[\begin{array}{ccc} a_{11} & \cdots & a_{1n}\\ \vdots & \ddots & \vdots \\ a_{m1} & \cdots & a_{mn} \end{array}\right]}_{m\times n}$$

### 2.2. Support Vector Machines

Given the set of training samples *D* = {(*x*_1_, *y*_1_), (*x*_2_, *y*_2_), …, (*x*_n_, *y*_n_)} *y_i_* ∊ {−1, 1}, the most basic objective of the SVM is finding a division hyperplane from the sample space according to the training samples D to separate data that have different characteristics. The following linear equation describes the hyperplane function [31]
(6)$$w^{T}x+b=0$$
where *w* = (*w*_1_, *w*_2_, …, *w_d_*) is the normal vector that determines the direction of the hyperplane, and *b* is a constant term that decides the distance between the hyperplane and the coordinate origin. Obviously, the divided hyperplane is constrained by *w* and *b*. The distance from point *x* belonging to the sample space to the plane (*w*,*b*) can be written as
(7)$$r=\frac{\left|w^{T}x+b\right|}{\left\|w\right\|}.$$

Suppose the plane (*w*,*b*) can classify the training samples correctly and has the following constraints for (*x_i_*, *y_i_*) ∊ *D*.
(8)$$\left\{ \begin{array}{l} w^{T}x_{i}+b\geq +1,y_{i}=+1\\ w^{T}x_{i}+b\leq +1,y_{i}=-1 \end{array}\right.$$

As shown in Figure 2, the training sample point that is nearest to the hyperplane and gives a true statement when substituted into Equations (2)–(7) is called the “support vector.” The sum of the distances of the two types of heterogeneous support vectors to the hyperplane is
(9)$$r=\frac{2}{\left\|w\right\|}.$$

The plane of the “maximum margin” is equivalent to finding the parameter *w* and *b*, which must satisfy Equation (7).
(10)$$\max_{w,b}\frac{2}{\left\|w\right\|}\;s.t.y_{i}(w^{T}x_{i}+b)\geq 1,\;i=1,2,3,\ldots ,m$$

Obviously, ${\left\|w\right\|}^{-1}$ should take the maximum value in order to maximize the interval. Therefore, Equations (2)–(9) is rewritten as
(11)$$\min_{w,b}\frac{1}{2}{\left\|w\right\|}^{2}.$$

## 3. Algorithm Simulation

To test the classification and recognition of the fault signal through the combined algorithms of ICA and SVM, MATLAB software was used to simulate three types of vibration signals, namely, the signal from the constant meshing gear $S_{1}\left(t\right)\prime$, the output gear pair signal $S_{2}\left(t\right)\prime$, and the signal from a broken tooth $S_{cp}\left(t\right).$ In order to simplify the simulation process, the other vibration sources in the gearbox were not considered when simulating the gearbox conditions. There were two different working conditions, which were convolutions of the three kinds of signals. Two hundred and forty sets of data were collected separately under different conditions. Two hundred groups of data were applied to train the classifier, and forty groups of data were used to test the accuracy of the classifier. The noise signals were replaced by vibration signals due to difficulty in the simulation of the noise signals and the correlation between noise and vibration. The description of the simulated signals is shown in Table 1.

The parameters of the simulation are consistent with the gearbox parameters to make the simulation closer to the actual situation. In this study, the gearbox of a loader from Shandong Lingong Construction Machinery Co., Ltd., that had three gears, was used, as shown in Figure 3. 

Aiming at the noise signal of gearbox failure, this paper designs a diagnostic algorithm based on ICA and SVM. The FastICA principle is used to separate the independent sources in the signal, and the correlation coefficients of the noise signal and the independent source are used to construct the eigenvector of the noise signal. The algorithm flow is shown in Figure 4. Gaussian kernel is used. The function SVM model trains the classifier to diagnose the fault of the gearbox. The algorithm flow is shown in Figure 5.

### 3.1. Simulation of the Original Signal

The second gear was selected, and the output speed of the torque converter was 1200 rpm in this working condition. The influence of the vibration signal of the torque converter was ignored.

In the simulation, the signal of the constant meshing gear $s_{1}\left(t\right)$ is as follows:(12)$$s_{1}(t)=\overset{n}{\underset{i=1}{\sum }}A_{i}\cos(2\pi if_{m1}t+\varphi_{i})$$
where $A_{i}$ and $\varphi_{i}$ are the amplitude and phases of the *i*th order vibration of the constant meshing gear pair, respectively, and $f_{m1}$ is the gear meshing frequency. 

In this paper, the frequency of the input shaft is $f_{n1}=20Hz$. The torque converter output speed is 1200 rpm. The gear meshing frequency is:(13)$$f_{m1}=20\times f_{n1}=400Hz.$$

Assuming the second-order vibration response of the gear pair is known, $A_{1}=0.5\;\mathrm{um}$, $A_{2}=1\;\mathrm{um}$, $\varphi_{1}=\varphi_{2}=0$. Then,
(14)$$s_{1}(t)=0.5\cos(800pt)+\cos(1600pt).$$

The output gear pair signal $s_{2}\left(t\right)$ is as follows:(15)$$s_{2}(t)=\overset{n}{\underset{i=1}{\sum }}B_{i}\cos(2\pi if_{m2}t+\theta_{i})$$
where $B_{i}$ and $\theta_{i}$ are the amplitude and phases of the *i*th order vibration of the output gear pair, respectively, and $f_{m2}$ is the frequency of the output gear pair in the second gear condition. 

Output shaft frequency $f_{n2}$ is:(16)$$f_{n3}=f_{n1}\times \frac{20}{52}\times \frac{60}{41}=11.3Hz.$$

The frequency of the output gear pair $f_{m2}$ is:(17)$$f_{m2}=41\times f_{n1}\times \frac{20}{52}\times \frac{60}{41}=462.5Hz.$$

In addition, the amplitudes $B_{1}=0.7\;\mathrm{um}$, $B_{2}=1.4\;\mathrm{um}$, and phases $\theta_{1}=\theta_{2}=0$.
(18)$$s_{2}(t)=0.7\cos(925\pi t)+1.4\cos(1850\pi t)$$

The faulty gear was selected from the driving wheel and the broken tooth fault was applied. A broken tooth in the gearbox will generate a local impact phenomenon, and the vibration signals show the impact given by Equations (3)–(5), where $s_{cp}\left(t\right)$ is defined as
(19)$$s_{cp}(t)=\overset{n}{\underset{i=1}{\sum }}\overset{m}{\underset{j=1}{\sum }}A_{cji}exp\left[\frac{-2\pi \varepsilon_{dj}}{\sqrt{1-\varepsilon_{dj}^{2}}}f_{dj}(t-\tau_{ci})\right]\cos\left[2\pi f_{dj}(t-\tau_{ci})\right]$$
where *n* is the number of shocks in the signal, *m* is the order of the natural frequency, and $A_{cji}$ is the amplitude of the *i*th impact when the machine has the jth order of natural frequency. To simplify the simulation process, $A_{cji}$ was assigned a random number from 0 to 10. The variables $\varepsilon_{dj}$ and $f_{dj}$ represent the jth order damping ratio and natural frequency, respectively; $\tau_{ci}$ is the moment of the ith impact, which is given by $\tau_{ci}=\left(i-1\right)T_{c}+\tau_{c0}$; $T_{c}$ is the cycle of the gear failure; and $\tau_{c0}$ is the initial time of the impact response, i.e., $\tau_{c0}=0.3T_{c}$. It was assumed that the two natural frequencies were 1100 and 3500 Hz. According to the actual working condition of the gearbox, $\varepsilon_{d1}=0.04$, $\varepsilon_{d2}=0.03$, $A_{c1}=A_{c2}\in \left(0,10\right)$, $\tau_{c1}=\tau_{c0}=0.3T_{c}=0.2\times 1/f_{n3}=0.027$, and $A_{c1}=A_{c2}=4\;\mathrm{um}$. The time and frequency diagrams of the three source signals are plotted in Figure 6.

### 3.2. Simulation of Working Condition Signals

Six data collectors were used in the simulation. Suppose that each source signal from two different channels reach the same collector. Therefore, the data from each collector can be expressed as
(20)$$x(t)=\left[x_{1}(t);\cdots ;x_{i}(t);\cdots ;x_{6}(t)\right](1<i<6)$$
where $x_{i}\left(t\right)$ is the data collected by the ith collector. The calculation method is as follows:(21)$$x\left(t\right)=b\left(i,:\right)[\begin{array}{c} s_{1-1}\left(\left(st_{i}\left(1\right)+1\right):\left(N+st_{i}\left(1\right)\right)\right)\\ s_{1-2}\left(\left(st_{i}\left(2\right)+1\right):\left(N+st_{i}\left(2\right)\right)\right)\\ s_{2-1}\left(\left(st_{i}\left(3\right)+1\right):\left(N+st_{i}\left(3\right)\right)\right)\\ s_{2-2}\left(\left(st_{i}\left(4\right)+1\right):\left(N+st_{i}\left(4\right)\right)\right)\\ s_{3-1}\left(\left(st_{i}\left(5\right)+1\right):\left(N+st_{i}\left(5\right)\right)\right)\\ s_{3-2}\left(\left(st_{i}\left(6\right)+1\right):\left(N+st_{i}\left(6\right)\right)\right) \end{array}]$$
where b is an arbitrary 6×6 mixed matrix, and N is the number of FFT calculations. A 0.2-s signal was collected at each collector to show the difference between the data collected by each collector. The vibration signal of the gearbox under the normal condition is shown in Figure 7. The fault signal corresponding to the broken gear from six data collectors is shown in Figure 8.

### 3.3. Feature Extraction

Due to the source signal emitted under the fault condition being generated by the convolution of three independent source signals in the simulation process, three independent components were selected to estimate the source signal. The FastICA algorithm was applied to separate the data for different conditions. First, the independent matrix was calculated. Second, the results of the signal were estimated by the matrix in the frequency domain, and the amplitude of the signal was corrected. Finally, the independent source was obtained through the iterative feedback tuning (IFT) criterion.

The results separated by FastICA are shown in Figure 9, and the three separated independent source signals have obvious periodic characteristics. By comparing the independent source signals in Figure 9 and the source signals in Figure 6, it can be seen that the separated signals have a certain degree of reduction of the normally meshing gear pair signal $s_{1}\left(t\right)$ and the output gear pair signal $s_{2}\left(t\right)$. Numerous differences in the sequence, amplitude, and shape can be found when comparing the results of the separation of the source signals. This can be attributed to the correlations between the signals used for the convolution. However, the independent components, once separated by FastICA, have obvious periodic characteristics and similarities when compared with the mixed source signals. When the fault signal was decomposed, the first signal contained the basic features of the fault signal components. The signals separated by FastICA are different and can be used to characterize the mixed signals under any condition.

### 3.4. Fault Identification

To improve the training precision, the correlation coefficients between the independent components and source data were used as the input parameters of the SVM classifier. The NIC*n* is the nth independent component extracted under normal conditions(n = 1,2,3), and the FIC*n* is the nth independent component extracted under fault conditions (n = 1,2,3). The procedure for the calculation of the correlation coefficient can be summarized as follows:

Step 1: Represent the different conditions by applying $S_{1}$, $S_{2},S_{3}$ according to the classification of the conditions. The FastICA algorithm is used to extract *m* independent components, which are recorded as ${\mathrm{IC}}_{i}$.

Step 2: Calculate the correlation coefficient between the first independent component of ${\mathrm{IC}}_{i}$ and each data channel of the signal, and take the absolute maximum. The first feature value $F_{1}$ is the sum of the *3* numbers.

Step 3: Repeat the second step to obtain the feature value $F_{2}$, $F_{3}$. The feature vectors are column vectors from $F_{1}$ to $F_{3}$.

Step 4: Repeat the second and third steps for the training signals, and obtain the feature vector of all signals.

The correlation coefficients of the NIC, FIC, and gearbox data collected under different conditions are shown in Table 2. It is worth noting that for n different conditions, only repeat step 2 and step 3 to get the eigenvectors of all signals.

The feature vector of each group is the sum of the correlation coefficients. Table 3 shows the feature vector under the two normal and two fault working conditions. It can be seen from the table that the data collected under normal working conditions are highly correlated with the characteristic signal extracted from normal signals by the FastICA algorithm, and the data collected under the faulty working conditions are highly correlated with the characteristic signal extracted from fault signals. This clear correlation indirectly proves that the FastICA algorithm successfully extracts the characteristic components of different signals.

Only one classifier can be trained, as there are only two types of working conditions in the set. The SVM was trained with approximately 160 groups of data from the two working conditions, and 40 samples were used to test the classifier performance. Each sample was a 2×1 matrix that could be expressed in two-dimensional coordinates. As shown in Figure 10, the two working conditions were clearly distinguished by a dividing hyperplane.

The normal and fault signals have obvious differences while calculating the feature vector. The normal signals are mainly underneath the division line because of the higher correlation coefficient between NIC and the normal conditions than between NIC and the fault conditions. The same reasoning can be applied to fault signals on the other side of the dividing line. Forty samples were tested, and there were only two false positives under the two abovementioned conditions. Therefore, the misdiagnosis rate was 2/40×100%=5%.

## 4. Noise Diagnosis Experiment of Gearbox Failure 

The fault situation during the process of gearbox operation is more complex than in the simulation not only because of the noise of a single component, but also because all kinds of aliasing noise and background noise are present. To verify the feasibility of the proposed algorithm in the noise diagnosis of the gearbox fault, the noise measurement platform was built to collect the data, and the algorithm was used to diagnose the failure of the gearbox.

### 4.1. Construction of Test Platform

Testing was performed in the gearbox laboratory of Shandong Lingong Construction Machinery Co., Ltd. A constant current source preamplifier (26CA) and free-field acoustic intensity sensor (46AE) from GRAS were applied. Before testing, the sound intensity sensor was first calibrated, then installed on the preamplifier and connected to the high-performance multichannel data acquisition system (LMS SCADAS) by BNC connector. Gear wear is the specific fault type considered in this paper, the length and depth of which was defined as 55 mm and 0.5 mm, respectively. The experiment was conducted using a test platform comprised of a motor, gearbox, speed sensor, constant current source preamplifier, free field acoustic intensity sensor, calibrator, and LMS SCADAS Mobile SCM05, as shown in Figure 11.

The Chinese national standard GB/T 3767-2016/ISO3744:2010 “Acoustics-Determination of sound power levels and sound energy levels of noise sources using sound pressure-engineering methods for an essentially free field over a reflecting plane” was referred to arrange the positions of 13 sound intensity sensors, as shown in Figure 10. First, the speed of the test rig was adjusted to 1200 r/min, and noise data under the normal conditions were collected when the speed sensor had measured the constant speed of the out shaft. The sampling frequency was 40,960 Hz, and the sampling time was 10 min. Next, the gearbox was stopped, the fault gear ring was replaced, and the signals corresponding to the fault conditions were collected.

### 4.2. Noise Spectrum Analysis

The two types of working noise data collected by the data acquisition system were divided into 120 groups with 5 s as the signal sampling period. There were 240 groups of samples, 200 groups of which were the training samples, and 40 samples were used to test the classification performance of the classifier. Signals were randomly intercepted to obtain 0.5 s signal fragments, as shown in Figure 12 and Figure 13.

Frequency spectrum analysis was performed for noise data from 13 measuring points under different working conditions, as shown in Figure 10 and Figure 11. The main frequency components of the noise collected under the normal working condition were 300.2, 900.5, 1247, and 1801 Hz. The speed measured by the velocimeter was 1317 r/min when the velocity was constant. Using the transmission ratio, the meshing frequency of the gear with 22 teeth was calculated to be approximately 300 Hz, the meshing frequency of the driving gear when in the second gear was calculated to be 900 Hz, and the meshing noise of the second-order fitting and of the engine were calculated to be 1800 and 1247 Hz, respectively. The results of the calculation are consistent with the spectral analysis components of the experimental data. From Figure 11, the main frequency components of noise acquisition under gearbox fault conditions are 300.5, 415, 900.5, 1247, and 1801 Hz. Among these, the frequency components of 300.5, 900.5, 1247, and 1801 Hz are the same as normal working conditions, so the noise at 415 Hz might be the frequency of the fault ring.

### 4.3. Independent Source Separation

The MDL criterion was used to calculate the noise of the two working conditions, using the seven independent sources. The FastICA algorithm in MATLAB was applied to separate the data. First, the matrices that were independent of each other were calculated. Second, the results of the signal were estimated using the matrix in the frequency domain, and the amplitude of the signal was corrected. Finally, the independent component of the time domain was obtained through the IFT criterion.

The time-frequency diagrams of the seven independent component signals separated from the normal working noise data are shown in Figure 14. All types of noise sources under the normal conditions were successfully separated and extracted by the FastICA algorithm. These mainly included the four above-mentioned principal frequencies: 300.5, 900.5, 1801, and 1247 Hz, as well as the approximate white Gaussian noise. In Figure 15, the algorithm separates four frequency components that correspond to normal conditions and the frequency of the gear ring fault.

### 4.4. Parameter Calculation of SVM

Two samples were extracted from the training set under different conditions. Calculation of the correlation coefficients was performed in the manner described in Section 3. The results are shown in Table 4 and Table 5.

The correlation coefficients between the independent components and the source signals from each working condition are obviously smaller than the results of the simulation because the signal has been separated into seven independent components. Clearly, the first five independent components provide the main characteristics of the source signal, while the remaining components can be regarded as the high-frequency noise of the source signal. This conclusion is made according to the calculation of the input parameters based on the SVM algorithm and the sum of the correlation coefficients for each row. There are no obvious gaps between the feature vectors, which may be due to the difference in gear wear under the two conditions and the masking of the vibration effect by random noise. 

One hundred groups of normal data samples and 100 groups of fault data samples were selected to train the SVM classifier. The division hyperplane of the SVM with the Gauss kernel function is plotted in Figure 16, in which it can be observed that the clusters corresponding to the samples for the two conditions are well separated. Forty groups of normal and fault signals were applied to verify the performance of the classifier, and there were three false positives. Therefore, the error probability was 3/40×100%=7.5%.

## 5. Conclusions

(1) A fault diagnosis algorithm based on ICA and SVM is proposed in this paper. The gearbox of a loader was taken as the research object, and source signals were used to correctly diagnose gearbox failure by applying feature extraction and classification recognition as the processing technologies. The signals collected from the gearbox, which contained the principal feature information of the fault, were extracted by the ICA algorithm. The correlation coefficient was used as the characteristic parameter of fault diagnosis, as it can improve the robustness of detection systems due to the high correlations between the independent components and working conditions.

(2) Three independent source signals in the gearbox were simulated using MATLAB, and the convolution hybrid model was used to generate the six channel signals under two working conditions. Two hundred groups of simulation data were collected. Two groups of samples were selected from each condition to display the entire process of signal processing. The training of the SVM was implemented using 160 groups of samples, and 40 groups were used to test the accuracy of the classifier. There were two false positives under the two conditions; therefore, the misdiagnosis rate was 5%. 

(3) A noise testing platform with 13 measurement points was built on the gearbox test rig according to the Chinese national standards, and two gear ring wear conditions were defined. One hundred and twenty groups of data under each condition were collected for the research. The algorithm established to calculate the correlation coefficient between the independent component and the source signal was applied to train the classifier and test its accuracy. There were three false positives; therefore, the error probability was 7.5%. 

(4) The experimental results show that the method proposed in this paper can effectively extract various mechanical fault features from signal data, providing significant engineering value. Furthermore, this research provides an innovative approach for the feature extraction of mechanical faults from signal data.

## Figures and Tables

**Figure 1 ijerph-16-04868-f001:**
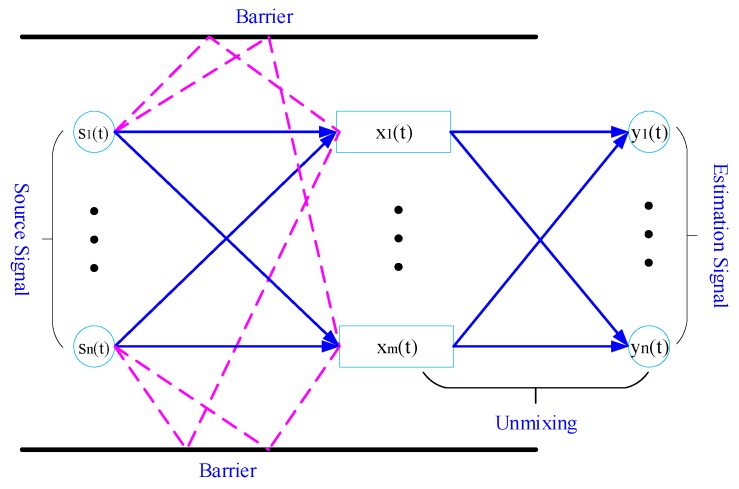
Linear convolution mixed model.

**Figure 2 ijerph-16-04868-f002:**
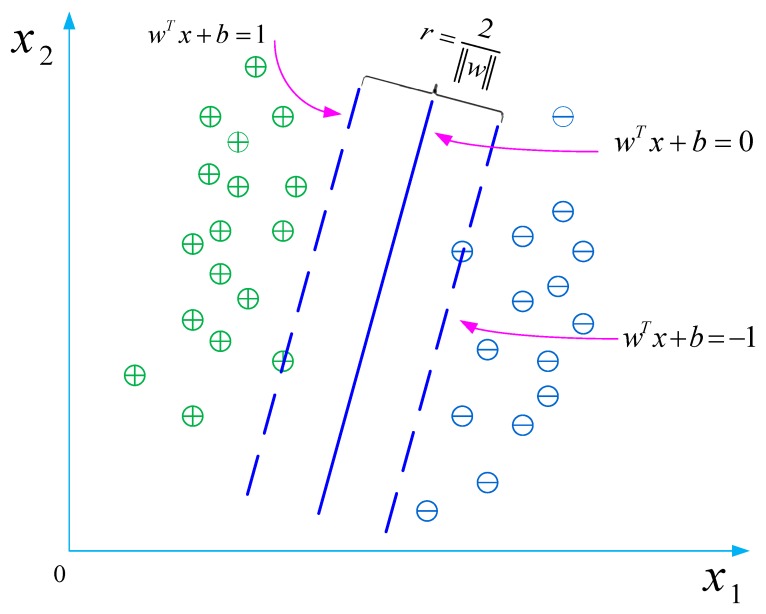
Principle of the support vector machine (SVM) classifier.

**Figure 3 ijerph-16-04868-f003:**
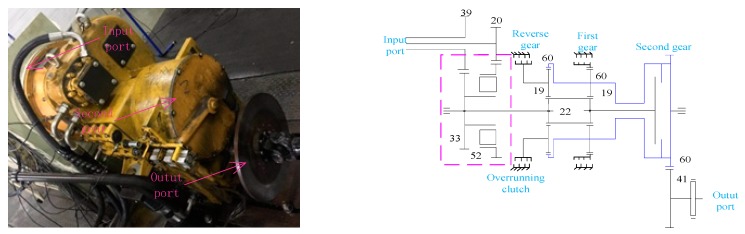
Transmission principle of loader gearbox.

**Figure 4 ijerph-16-04868-f004:**
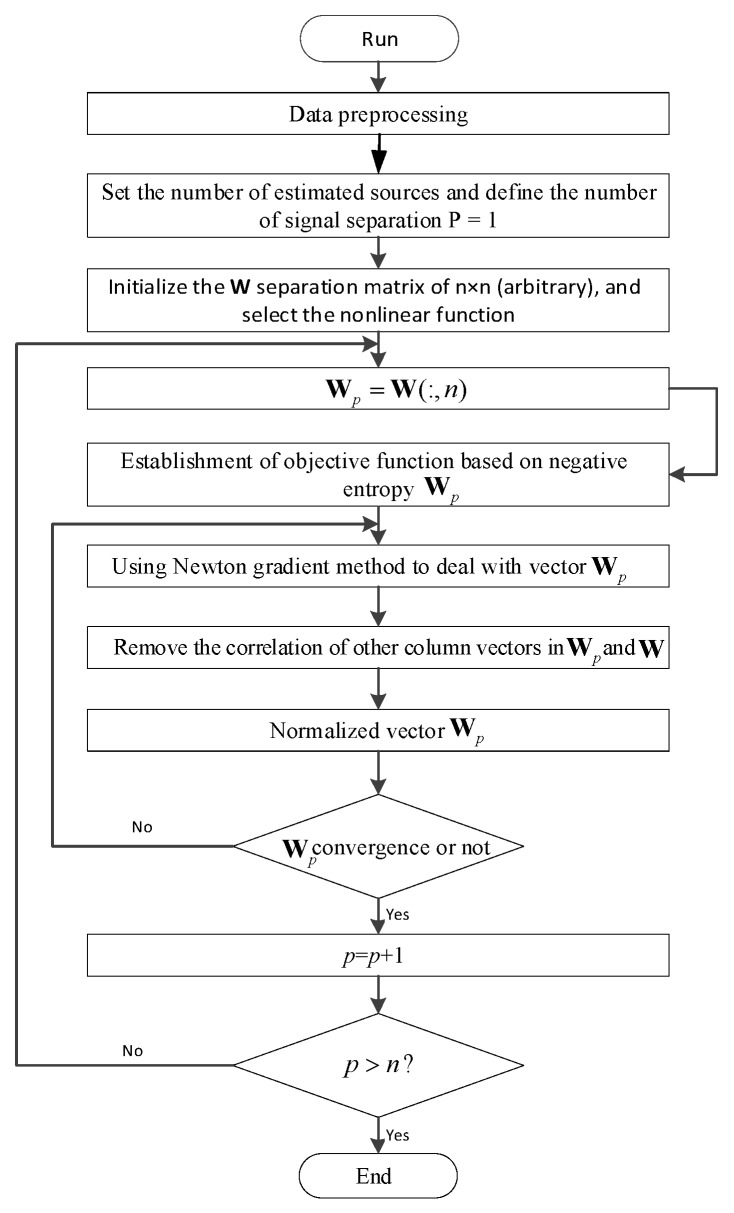
FastICA algorithm flow.

**Figure 5 ijerph-16-04868-f005:**
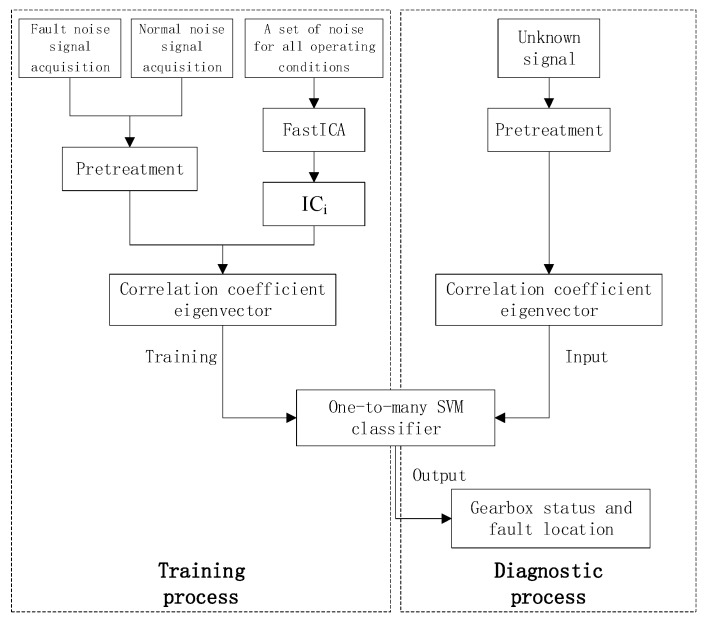
Independent component analysis (ICA)-SVM-based diagnostic algorithm flow.

**Figure 6 ijerph-16-04868-f006:**
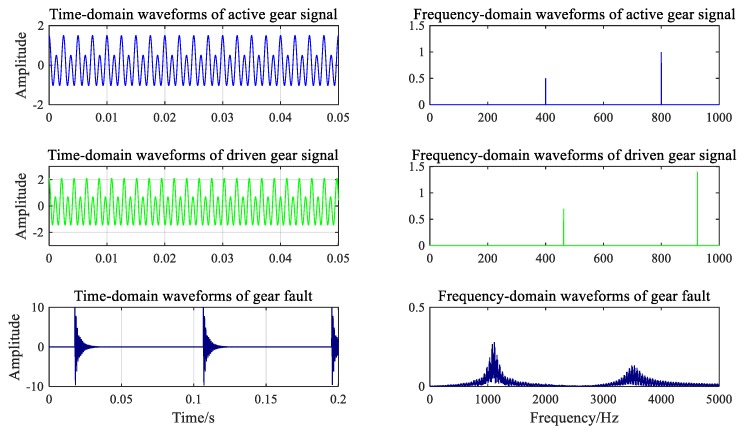
Time and frequency diagrams of three signals.

**Figure 7 ijerph-16-04868-f007:**
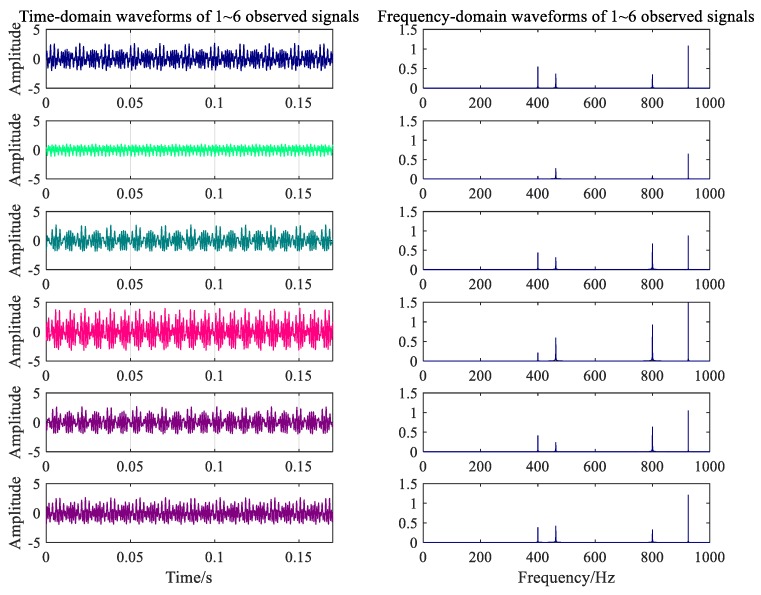
Time-frequency waveforms of signals under normal conditions.

**Figure 8 ijerph-16-04868-f008:**
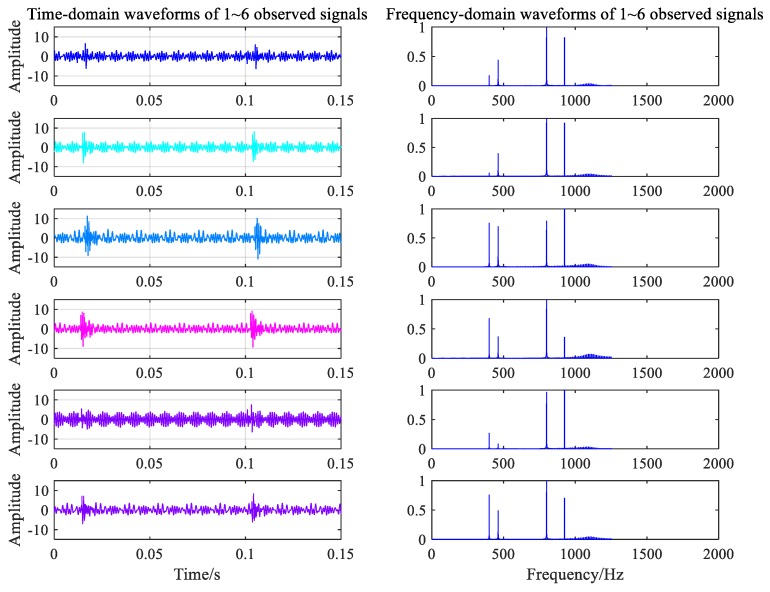
Time-frequency waveforms of signals under fault condition.

**Figure 9 ijerph-16-04868-f009:**
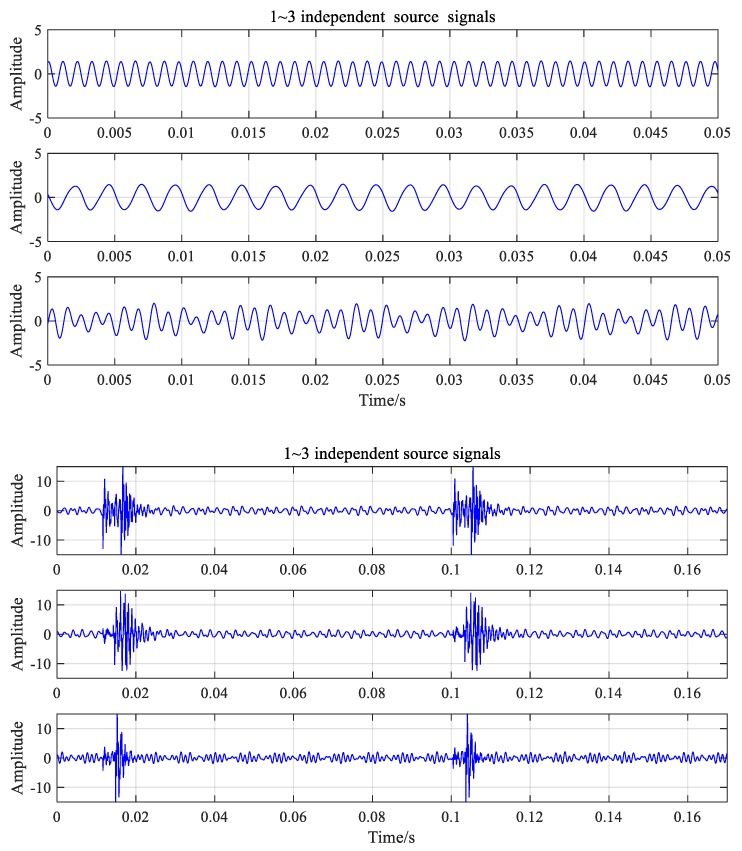
Separation results by FastICA.

**Figure 10 ijerph-16-04868-f010:**
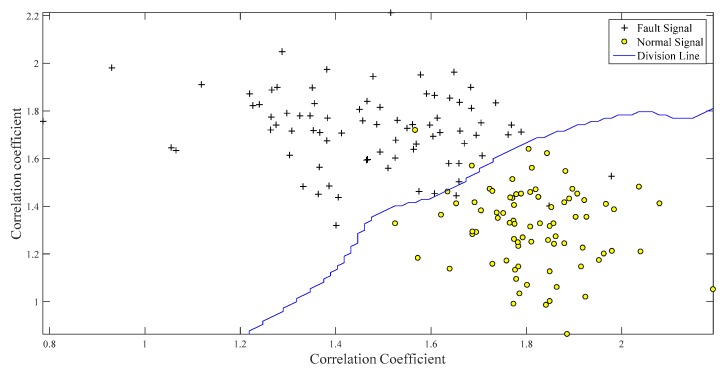
Characteristic distributions of samples under normal and fault conditions.

**Figure 11 ijerph-16-04868-f011:**
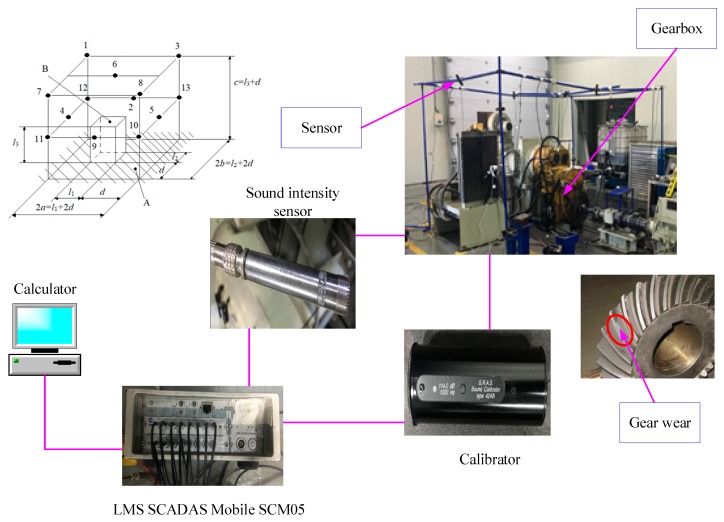
Construction of signal acquisition platform. Black solid point represents the sound intensity sensor, A represents reflector, B represents datum body, 2a represents measuring face length, 2b represents measuring face width, c represents height of measuring face, d represent measuring distance (80 cm), $l_{1}$ represents datum length (80 cm), $l_{2}$ represents datum width (60 cm), $l_{3}$ represents height of datum (120 cm).

**Figure 12 ijerph-16-04868-f012:**
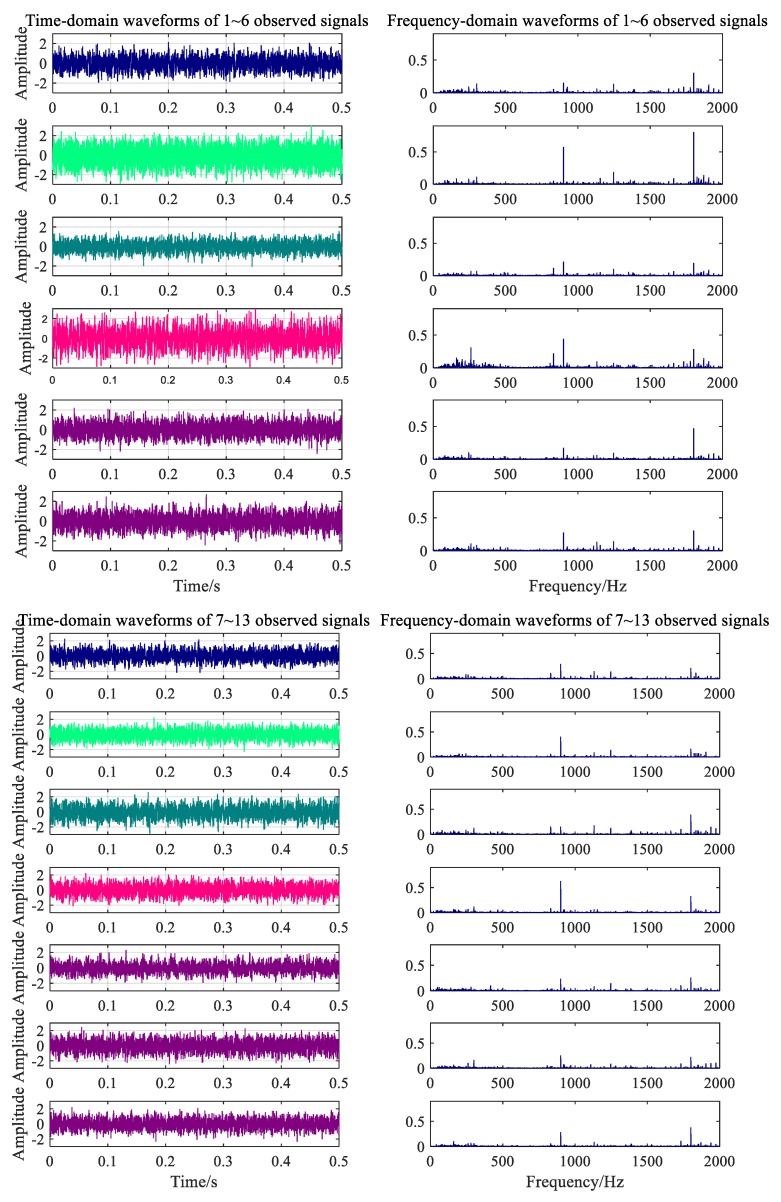
Time-frequency waveforms of measured signal under normal condition.

**Figure 13 ijerph-16-04868-f013:**
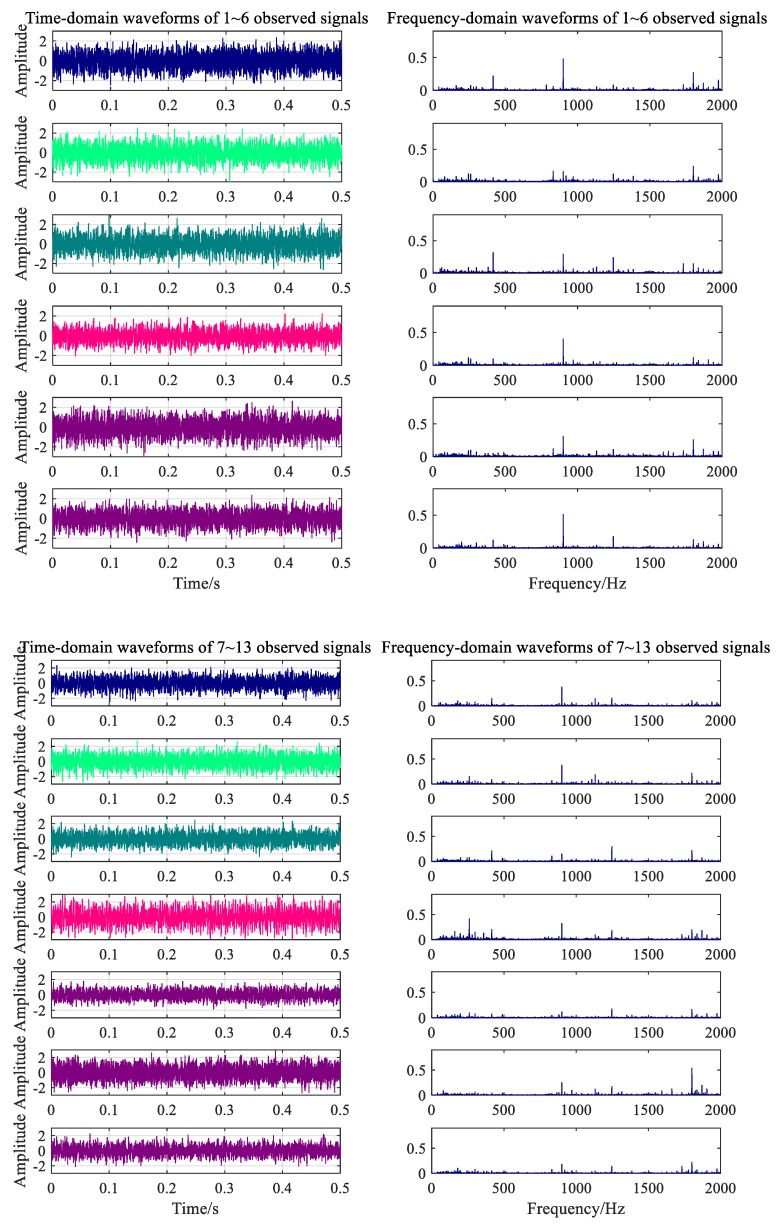
Time-frequency waveforms of measured signal under fault condition.

**Figure 14 ijerph-16-04868-f014:**
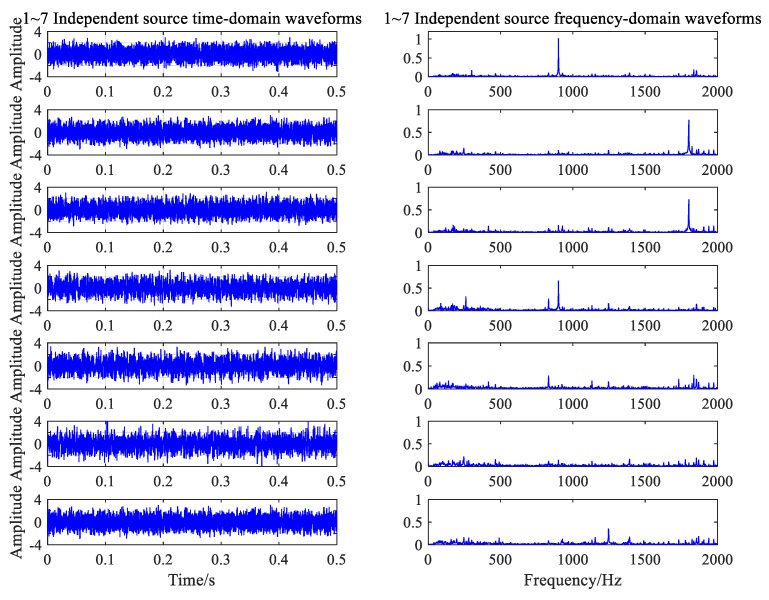
Independent components of the normal condition.

**Figure 15 ijerph-16-04868-f015:**
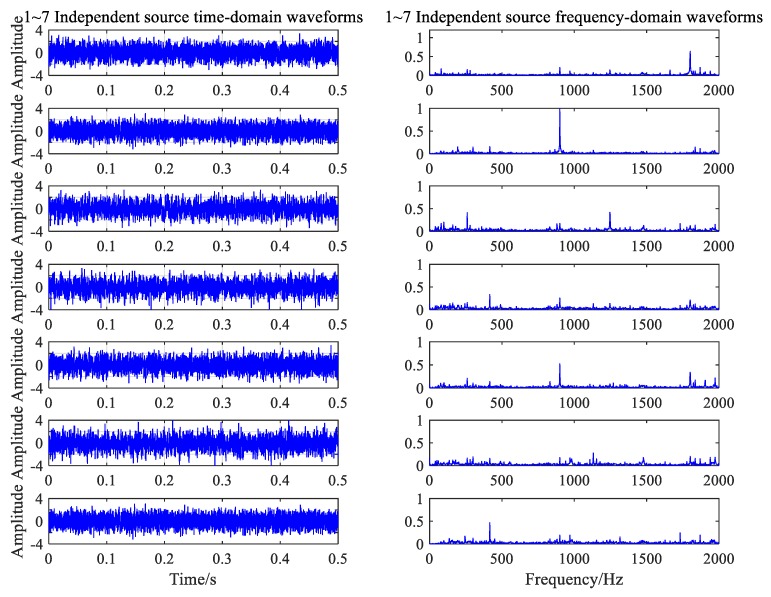
Independent components of the fault condition.

**Figure 16 ijerph-16-04868-f016:**
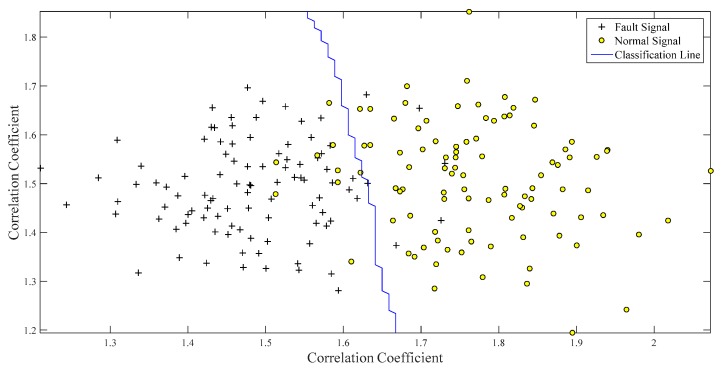
Classification of two classes using SVM.

**Table 1 ijerph-16-04868-t001:** Description of simulation signals.

Signals	Expression
Normal	$s_{1}\left(t\right)*s_{2}\left(t\right)$
Fault	$s_{1}\left(t\right)*s_{2}\left(t\right)*s_{cp}\left(t\right)$

**Table 2 ijerph-16-04868-t002:** Correlation coefficients of the independent component and measured source signal.

Working Condition	NIC1	NIC2	NIC3	FIC1	FIC2	FIC3
Normal1	0.5464	0.8076	−0.7904	0.1378	−0.2303	−0.4007
Normal2	0.5008	0.5905	−0.8645	0.3234	−0.3088	−0.2871
Fault1	0.4303	0.3140	−0.5269	−0.5054	0.7256	0.2977
Fault2	0.3467	−0.3878	−0.4345	−0.5601	−0.5010	−0.4469

**Table 3 ijerph-16-04868-t003:** Feature vector of each group.

Normal1	Normal2	Fault1	Fault2
$\left(\begin{array}{l} 2.1444\\ 0.7688 \end{array}\right)$	$\left(\begin{array}{l} 1.9558\\ 0.9193 \end{array}\right)$	$\left(\begin{array}{l} 1.2712\\ 1.5287 \end{array}\right)$	$\left(\begin{array}{l} 1.1690\\ 1.5080 \end{array}\right)$

**Table 4 ijerph-16-04868-t004:** Correlation coefficients of independent components from normal condition and source signals.

	NIC1	NIC2	NIC3	NIC4	NIC5	NIC6	NIC7
Normal1	−0.4457	0.3786	0.3969	0.2652	−0.0838	0.1085	−0.1083
Normal2	0.4512	0.3218	−0.3612	−0.2755	−0.0655	−0.0609	−0.1400
Fault1	0.4594	0.2638	0.2603	−0.2019	0.0870	0.0615	−0.0734
Fault2	−0.3279	−0.3823	−0.2488	−0.2905	−0.0593	0.0569	−0.0725

**Table 5 ijerph-16-04868-t005:** Correlation coefficients of independent components from fault condition and source signals.

	FIC1	FIC2	FIC3	FIC4	FIC5	FIC6	FIC7
Normal1	0.3504	−0.441	0.1837	0.0978	−0.2111	−0.1088	−0.0693
Normal2	0.3603	0.2635	−0.1372	−0.1918	0.3860	0.0649	−0.0848
Fault1	0.3942	0.3173	−0.1187	0.1528	0.2563	0.0946	−0.0937
Fault2	−0.3380	−0.4281	0.1407	0.1862	−0.3218	0.1344	0.1187

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
