# Peer review of "Fault Diagnosis of Loader Gearbox Based on an ICA and SVM Algorithm"

_ijerph, 2019, doi:10.3390/ijerph16234868_

Round 1

Reviewer 1 Report

-please add photo of measurements/applications (add arrows what is what)
-please put block diagram of proposed analysis/research/method
-please put labels to figures, SI units for example "Amplitude of what?"
-compare your methods with MSAF-RATIO-27-MULTIEXPANDED-4-GROUPS, method of selection ofamplitudes of frequencies (MSAF-5), shortened method of frequencies selection (SMoFS-10) and the obtained results of SVM.
-cite references from 2015-2019 Web of Science

for example about SVM, PCA

Recognition of acoustic signals of induction motor using FFT, SMOFS-10 and LSVM
EKSPLOATACJA I NIEZAWODNOSC-MAINTENANCE AND RELIABILITY
Volume: 17
Issue: 4
Pages: 569-574
DOI: 10.17531/ein.2015.4.12
Published: 2015

A PCA and Two-Stage Bayesian Sensor Fusion Approach for Diagnosing Electrical and Mechanical Faults in Induction Motors
IEEE TRANSACTIONS ON INDUSTRIAL ELECTRONICS
Volume: 66
Issue: 12
Pages: 9510-9520
DOI: 10.1109/TIE.2019.2891453
Published: DEC 2019

Wind Power Prediction Based on LS-SVM Model with Error Correction
ADVANCES IN ELECTRICAL AND COMPUTER ENGINEERING
Volume: 17
Issue: 1
Pages: 3-8
DOI: 10.4316/AECE.2017.01001
Published: 2017

Detection and identification of windmill bearing faults using a one-class support vector machine (SVM)
MEASUREMENT
Volume: 137
Pages: 287-301
DOI: 10.1016/j.measurement.2019.01.020
Published: APR 2019

Author Response

Dear Editors and Reviewers:

Thank you for your letter and for the reviewers’ comments concerning our manuscript entitled “Fault Diagnosis of Loader Gearbox based on an ICA and SVM Algorithm” (Manuscript ID: ijerph-647868). Those comments are all valuable and very helpful for revising and improving our paper, as well as the important guiding significance to our researches. We have studied comments carefully and have made correction which we hope meet with approval. Revised portion are marked in red in the paper. The main corrections in the paper and the responds to the reviewer’s comments are as flowing:

Point 1: please add photo of measurements/applications (add arrows what is what). 

Response 1: We are very sorry for our negligence of not adding photo of measurements/applications. We have re-written Picture 3 and added arrows and labels according to the Reviewer’s suggestion.

Point 2: please put block diagram of proposed analysis/research/method.

Response 2: It is really true as Reviewer suggested that we need to put the ICA-SVM-based diagnostic algorithm flow in the article. So, we have added Picture 4 to introduce analysis/research/method of the entire article.

Point 3: please put labels to figures, SI units for example "Amplitude of what?".

Response 3: Thank you very much for finding out the problem about the error of amplitude unit. We have made correction according to the Reviewer’s comments.

Point 4: compare your methods with MSAF-RATIO-27-MULTIEXPANDED-4-GROUPS, method of selection of amplitudes of frequencies (MSAF-5), shortened method of frequencies selection (SMoFS-10) and the obtained results of SVM.

Response 4Thank you very much for your comments and suggestions. MSAF-RATIO-27-MULTIEXPANDED-4-GROUPS provided by reviewer is a very worthy method. This method makes use of the difference of FFT spectrum to find the possibility of solving the problem which have not solve in time domain. The method of this paper focuses on the separation of noise variables, hoping to get independent fault components. We revised the article and added picture 5, hoping to express our method more clearly.

Point 5: cite references from 2015-2019 Web of Science.

Response 5It is really true as Reviewer suggested that the citations on SVM and PAC in this article are not comprehensive enough. Considering the Reviewer’s suggestion, we have added several recent articles on SVM and PAC, such as [26], [27], [28], [29].

We tried our best to improve the manuscript and made some changes in the manuscript.  These changes will not influence the content and framework of the paper. And here we did not list the changes but marked in red in revised paper.

Once again, thank you very much for your comments and suggestions.

Reviewer 2 Report

Manuscript Number:  ijerph-647868

Title:  Fault Diagnosis of Loader Gearbox based on an ICA and SVM Algorithm

Type: Article

Comments

This paper presents a work on implementation of two algorithms ICA and SVM for Fault Diagnosis of Loader Gearbox.

The authors first present a state of the art on the application of the ICA algorithm (bioengineering, communication, speech recognition and fault diagnosis) and its variants. According to the authors, SVM has high computational complexity and low accuracy in direct classification using raw data with redundant characteristics. Both ICA and SVM algorithms are presented in the article. The authors propose using the ICA technique for extracting characteristics from the original data and to reduce correlation between signals. The correlation coefficients between the feature vector and the source data were then used as input parameters of the SVM classifier. The authors explain the process, the authors then present a numerical validation of the proposed method and an experimental validation. The results obtained appear very promising. Nevertheless, the authors should clearly explain the process. There are still many outstanding issues to be dealt:

Questions and Comments:

Equation (3) should be reviewed. In the frequency domain, it is a classic product and not a convolution product. In the simulation, the noise signals have been replaced by vibration signals because of difficulty in simulation of the noise signals and the correlation between the noise and the vibrations. The relationship between fm1 and fn1 should be explained. It should be explained why equation (12) is obtained from Equation (11). Same for equation (14); Why fm1 = 800 Hz and fm2 = 925 Hz? Clarify if in equation (17), it is a convolution or classic product. In section 2.1 (Independent component analysis) b (t) represents the mixing matrix and in the simulation m (t) represents the mixing matrix. The same notations should be used. Does Figure 4 represent the 3 source signals? The signal of the constant meshing gear ?1 (?), the output gear pair signal ?2 (?) and the waveform of gear fault. After applying FastICA algorithm, Figure 7 shows the three independent sources. Why these three sources are different from the three sources in Figure 4? Figure 7 should be explained with respect to Figure 4. Section 3.4 should be explained:

In section 3 .4 “Fault identification”, the authors talk about of n independent sources whereas there are only 3 sources in the simulation. This makes it difficult to understand the algorithm. We should keep the 3 sources in this section and then make a generalization for n independent sources.

The NICn is the nth independent component extracted under normal conditions and the FICn is the nth independent component extracted under fault conditions. Why in Table 2, in the first line (Normal 1) the FICn exists and different from zero? Or I did not understand the proposed approach.

The results obtained in section 4 "Noise diagnosis experiment of gearbox failure" are very interesting. Authors should explain how they got the source signals to calculate NICn and FICn

The work done by the authors corresponds to the current problem of tracking defects in rotating machines and as part of predictive maintenance. This work has the merit of proposing a new default detection method. The method has been experimentally validated. This method appears promising. This article could be published in the journal if the authors provide answers to all these questions.

Author Response

Dear Editors and Reviewers:

Thank you for your letter and for the reviewers’ comments concerning our manuscript entitled “Fault Diagnosis of Loader Gearbox based on an ICA and SVM Algorithm” (Manuscript ID: ijerph-647868). Those comments are all valuable and very helpful for revising and improving our paper, as well as the important guiding significance to our researches. We have studied comments carefully and have made correction which we hope meet with approval. Revised portion are marked in red in the paper. The main corrections in the paper and the responds to the reviewer’s comments are as flowing:

Point 1: Equation (3) should be reviewed. In the frequency domain, it is a classic product and not a convolution product.  

Response 1: We are very sorry for our negligence of unclear explanation of Equation (3). We have re-written the annotation of formula 3 and removed confusing sentences.

Point 2: In the simulation, the noise signals have been replaced by vibration signals because of difficulty in simulation of the noise signals and the correlation between the noise and the vibrations. The relationship between fm1 and fn1 should be explained. It should be explained why equation (12) is obtained from Equation (11). Same for equation (14); Why fm1 = 800 Hz and fm2 = 925 Hz?

Response 2: It is really true as reviewer suggested that we need to make equation (12) and equation (14) easier to understand. So we have added equation (13)、equation (16) and equation (17) to explain this question.

Point 3: Clarify if in equation (17), it is a convolution or classic product. In section 2.1 (Independent component analysis) b (t) represents the mixing matrix and in the simulation m (t) represents the mixing matrix. The same notations should be used.

Response 3: Thank you very much for finding out the disunity of expression of mixed matrix. We have made correction according to the reviewer’s comments.

Point 4: Does Figure 4 represent the 3 source signals? The signal of the constant meshing gear s1 (t), the output gear pair signal s2 (t) and the waveform of gear fault. After applying FastICA algorithm, Figure 7 shows the three independent sources. Why these three sources are different from the three sources in Figure 4? Figure 7 should be explained with respect to Figure 4.

Response 4Thank you very much for your comments and suggestions. As reviewer suggested that we've changed the description of picture 6(original picture 4) and added the description of comparison between picture 9 (original picture 7) and picture 6(original picture 4).

Point 5: In section 3 .4 “Fault identification”, the authors talk about of n independent sources whereas there are only 3 sources in the simulation. This makes it difficult to understand the algorithm. We should keep the 3 sources in this section and then make a generalization for n independent sources. The NICn is the nth independent component extracted under normal conditions and the FICn is the nth independent component extracted under fault conditions. Why in Table 2, in the first line (Normal 1) the FICn exists and different from zero? Or I did not understand the proposed approach.

Response 5Again thanks for pointing out the problem in section 3.4 “Fault identification”. We have made correction according to the reviewer’s comments. In fact, we have carried out experiments on four or more other sources, and we think the way of this paper can be expanded. For the problem in Table 2, the correlation coefficient in this paper is a statistical indicator indicating the linear correlation degree between variables. The larger the absolute value of the correlation coefficient is, the closer the linear relationship between the two variables is. Because of the uncertain interference problem, even the independent source separated by the way of FastICA, in the first line (Normal 1) the FICn cannot be 0.

Point 6: The results obtained in section 4 "Noise diagnosis experiment of gearbox failure" are very interesting. Authors should explain how they got the source signals to calculate NICn and FICn.

Response 6As reviewer suggested that we need to explain how to get the source signals. So, we have re-written Picture 11 and added a description of the sensor location.

We tried our best to improve the manuscript and made some changes in the manuscript.  These changes will not influence the content and framework of the paper. And here we did not list the changes but marked in red in revised paper.

Once again, thank you very much for your comments and suggestions.

Reviewer 3 Report

Gears are power transmissions widely used in gearboxes of different machines: automobiles, machine tools etc. The main failure mechanisms in gears are usually abrasive and corrosive wear, and pitting in the case of nonhardened gears. The predictive diagnosis of gears allows to plan the maintenance of gearboxes in a convenient way.

In this paper, the diagnosis of a loader's gearbox is realized by using multiple acoustic sensors to acquire signals and a combined algorithm based on independent component analysis (ICA) for features extraction and support vector machine classifier (SVM). Finally, the SVM decides the normal or faulty state of the gearbox on the base of previous training with specific inputs. The proposed model is tested by both theoretically generated and experimentally acquired signals, the prediction errors being 5% and 7% respectively.

The ICA-SVM technique was already used by other scientists to diagnose various mechanical systems (supplementary references 1 and 2). I appreciate the thorough description of the adopted model and the clarity of the presentation of the obtained results, but I still have some minor suggestions for the authors.

1. In my opinion, in both theoretical and experimental cases the prediction errors are too high for an SVM algorithm. This could be due to limited number of input signals used to train and test the SVM in the case of generated signals, or due to the overtraining in the experimental case. Did the authors tried to use half of data for training and half for testing?! I invite the authors to discuss more on this subject and to argue the obtained large prediction error of SVM. Also, they should compare the performances of their algorithm against those presented in literature.  

2. The first phrase of Abstract section must be reformulated, as gears do not fail due to the complicated structure of a loader and correlation between its parts.

3. Some constants are not defined: p from equation (1), and as(f) in equation (4). In equation (8) γ is r from Figure 2?!

Supplementary references:

1. https://doi.org/10.4028/www.scientific.net/amm.740.523

2. https://doi.org/10.1109/wcica.2006.1714144

Author Response

Dear Editors and Reviewers:

Thank you for your letter and for the reviewers’ comments concerning our manuscript entitled “Fault Diagnosis of Loader Gearbox based on an ICA and SVM Algorithm” (Manuscript ID: ijerph-647868). Those comments are all valuable and very helpful for revising and improving our paper, as well as the important guiding significance to our researches. We have studied comments carefully and have made correction which we hope meet with approval. Revised portion are marked in red in the paper. The main corrections in the paper and the responds to the reviewer’s comments are as flowing:

Response 1: Thank you very much for your comments and suggestions. According to the reviewer's suggestions, we conducted the following experiments:

When choosing a Gaussian kernel as the kernel function of SVM, let . The value of (g,C) uses the value of (50,1). g is taken as 2-4、2-3、2-2、2-1、20、21、22、23、24、25、26、27.After setting the parameters, a classification comparison experiment is performed. The best detection accuracy is 96.2% when g=2-4.

The penalty factor C represents the tolerance of the SVM classifier to losses, where g = 50 and C=(2-7、2-6、2-5、2-4、2-3、2-2、2-1、20、21、22、23、24). After setting the parameters, a classification comparison experiment is performed. The accuracy of the model does not change when g=(2-7:20) and can be maintained at 87.39%. Compared with the parameters of the kernel function, the penalty factor has less influence on the model.

The grid search method is used to exhaust the combination of two parameters, and the optimal model parameter combination is selected according to the five-fold cross-validation method. When the values of (g, C) are (0.015, 0.125), the accuracy of the classifier is the highest, and the accuracy is 97.6%.

We're sorry that the image cannot be uploaded due to software modification restrictions.The effect of the experimental simulation is better than the effect applied to the field. The problems may be more complicated, and we need to conduct further research. Many thanks to the reviewers for providing a feasible idea for our next research.

Point 2: The first phrase of Abstract section must be reformulated, as gears do not fail due to the complicated structure of a loader and correlation between its parts.

Response 2: Thank you very much for your comments on the abstract. We have made correction according to the reviewer’s comments.

Point 3: Some constants are not defined: p from equation (1), and as(f) in equation (4). In equation (8) γ is r from Figure 2?!

Response 3: We are very sorry for our negligence of equation (8). We have re-written equation (8).

We tried our best to improve the manuscript and made some changes in the manuscript.  These changes will not influence the content and framework of the paper. And here we did not list the changes but marked in red in revised paper.

Once again, thank you very much for your comments and suggestions.

Round 2

Reviewer 1 Report

the paper is good enough to publish

This manuscript is a resubmission of an earlier submission. The following is a list of the peer review reports and author responses from that submission.